# Conversion of D-Fructose into Ethyl Lactate Over a Supported SnO₂–ZnO/Al₂O₃ Catalyst

**Svitlana V. Prudius \*, Nataliia L. Hes and Volodymyr V. Brei**

Institute for Sorption and Problems of Endoecology of National Academy of Sciences of Ukraine, 13 General Naumov Str., Kyiv 03164, Ukraine; natalya2938@gmail.com (N.L.H.); brei@ukr.net (V.V.B.)

**\*** Correspondence: svitprud@gmail.com; Tel.: +38-067-7678855

**Abstract:** The study is directed to the search of a simple effective catalyst for ethyl lactate obtained from fructose as renewable raw material. A series of $SnO_2$-containing oxides prepared by impregnation of alumina were characterized by several techniques in order to determine their textural and acid-base properties. The transformation of a 13% fructose solution in 98% ethanol over $SnO_2/Al_2O_3$ catalysts using autoclave rotated with 60 rpm at 160 °C for 3 h was studied. It was found that doping $SnO_2/Al_2O_3$ samples with ZnO improves selectivity towards ethyl lactate. The supported $SnO_2$-$ZnO/Al_2O_3$ catalyst provides 100% fructose conversion with 55% yield of ethyl lactate at 160 °C. A possible scheme of fructose transformation into ethyl lactate on L-acid $^{IV}Sn^{4+}$ sites is discussed.

**Keywords:** fructose; ethyl lactate; $SnO_2/Al_2O_3$ catalyst

## 1. Introduction

Ethyl lactate obtained from lactic acid and ethanol is used mainly as green solvent, as well as in the food, cosmetic and pharmaceutical industries [1–3]. At present, ethyl lactate is considered as material for synthesis of monomeric lactide and ethyl acrylate [3]. Therefore, considerable attention is paid to the search for alternative methods for its preparation [4–8]. Some progress was made in the synthesis of alkyl lactates from dihydroxyacetone [4–6] and glycerol [7]. In a particular, the direct synthesis of ethyl lactate from glycerol solution in ethanol on $CeO_2/Al_2O_3$ catalyst at 230 °C was realized [7]. Recently, many works have been devoted to studying the activity of Sn-containing oxides, such as mesoporous MCM-41 [9,10], SBA-15 [10] materials and MFI [10], BEA [10], MWW [11] zeolites as catalysts for conversion of carbohydrates into alkyl lactates.

Holm with co-workers [8] found in 2010 that Sn-Beta zeolite directly catalyzes the transformation of mono- and disaccharides in methanol (0.1 mmol/L) into methyl lactate at 160 °C. Also, authors [12] showed that Zn-Sn-Beta converts sucrose into lactic acid with 54% yield at 190 °C after 2 h.

Numerous studies with different zeolites and their acidity showed that a combination of Lewis and Bronsted acidity catalyzes the selective conversion of monosaccharides to lactates [8,9]. It is known that Brønsted strong acid sites catalyze alternative reactions that lead to the formation of undesirable by–products [8–11]. Recent studies have shown that the addition of alkali ions to the reaction mixture or using dual metal tin-containing zeolite system as a catalyst increases the conversion of sucrose to lactic acid or lactate [12,13]. However, the preparation of Sn-zeolites is rather complicated procedure. In this work, we have used several Sn-containing oxides obtained by a simple impregnation method as catalysts of one-pot conversion of fructose into ethyl lactate.

## 2. Materials and Methods

Pure $SnO_2$ was prepared by precipitation from solution of $SnCl_4 \cdot 5H_2O$ with urea at 90 °C. After washing and drying at 110 °C, the precipitate was granulated and calcined at 550 °C in flowing air for 2 h.

Alumina-supported Sn-catalysts containing 10–25 wt.% $SnO_2$ were prepared by incipient wetness impregnation of commercial $\gamma$-$Al_2O_3$ (Alvigo) with calculated aqueous solutions of $SnCl_4 \cdot 5H_2O$ (Aldrich, 98%). Before the impregnation procedures, $\gamma$-$Al_2O_3$ was treated at 250 °C in flowing air for 3 h. The samples were denoted as $zSnO_2/Al_2O_3$, where $z$ is the $SnO_2$ content expressed in wt.%. The supported $SnO_2/Al_2O_3$ precursors were calcined at 550 °C for 2 h. Sample $10SnO_2/Al_2O_3$ was additionally impregnated with an aqueous solution of $Zn(CH_3COO)_2$ by incipient wetness impregnation. After thermal treatment, the zinc oxide content was 5 wt.%. The sample was denoted as $10SnO_2$–$5ZnO/Al_2O_3$.

Thermal analysis studies were carried out using a Perkin Elmer thermogravimetric analyser (TGA). Samples were purged with either nitrogen or oxygen and heated from room temperature to 800 °C at a heating rate of 10 °C/min.

XRD patterns of samples were recorded with DRON-4-07 diffractometer (CuK$\alpha$ radiation). Diffraction patterns were identified by comparing with those from the JCPDS (Joint Committee of Powder Diffraction Standards) database.

The textural parameters of the samples were calculated from the adsorption–desorption isotherms of nitrogen using the BET method (Quantachrome Nova 2200e Surface Area and Pore Size Analyser). The computational error for $S_{sp}$ determination is $\leq$2%.

UV-*Vis* diffuse reflectans spectra were recorded using a Perkin Elmer Lambda 40 spectrophotometer equipped with a diffuse reflectance chamber and integrating sphere (Labsphere RSA-PE-20). Prior to each experiment, samples were compacted in a sample holder to obtain a sample thickness of $\approx$2 mm. In order to determine the forbidden gap energy $E_0$, the reflectance spectra were recalculated to absorption spectra using Kubelka–Munk formula, $F = (h\nu(1 - R)^2/2R)^{1/2}$. $E_0$ values were determined from the nearly linear long-wave segment of absorption band plot extrapolated to interception with abscissa [14]. The material used as a reference was MgO.

Total number of acid or base sites was determined by reverse titration using n-butylamine or 2,4-dinitrophenol solution in cyclohexane, respectively, with bromthymol blue as an indicator. The highest acid or base strength was examined by the Hammett indicators method [15] using 0.1% solution of the corresponding indicators in cyclohexane.

For catalytic experiments, 13 wt.% solution of D-fructose (N98%, Merck) in anhydrous ethanol was used as a reaction mixture. The experiments were carried out in a rotated autoclave (60 rpm) at 160 °C for 3 h. Usually, 1.5 g of fructose, 10 g of anhydrous ethanol and 0.68 g (6 wt.%) of a catalyst were placed into a 25-mL Teflon can. The formation of insoluble products at these conditions was not observed.

The reaction products were analyzed using $^{13}C$ NMR spectroscopy (Bruker Avance 400, Karlsruhe, Germany). The conversion values of fructose (X) and selectivity of products (S, mol%) were calculated from $^{13}C$ NMR spectra. Yields were calculated as Y = S·X.

## 3. Results and Discussion

All synthesized samples were analyzed by several techniques in order to study their chemical, structural, textural and acid properties. Textural and acidic properties of synthesized $SnO_2/Al_2O_3$ oxides are summarized in Table 1.

**Table 1.** Textural parameters and total acidity of synthesized $SnO_2/Al_2O_3$ samples.

| Sample | Specific Surface Area, $m^2/g$ | Pore Volume, $cm^3/g$ | Average Pore Diameter, nm | Highest Acid Strength, $H_{0max}$ | Total Content of Acid Sites, mmol/g |
|---|---|---|---|---|---|
| $Al_2O_3$ | 290 | 0.82 | 10.7 | +3.3 | 1.2 |
| $SnO_2$ | 39 | 0.07 | 7.2 | +1.5 | 0.4 |
| $10SnO_2/Al_2O_3$ | 250 | 0.67 | 11.4 | +1.5 | 1.3 |
| $20SnO_2/Al_2O_3$ | 228 | 0.62 | 10.8 | +1.5 | 1.5 |
| $25SnO_2/Al_2O_3$ | 242 | 0.62 | 10.2 | +1.5 | 1.8 |
| $10SnO_2–5ZnO/Al_2O_3$ | 232 | 0.65 | 10.2 | +3.3 | 1.3 |

According to the thermogravimetric analysis of as-calcined $20SnO_2/Al_2O_3$ sample (Figure 1), the condensation of structural OH-groups with water release is observed at 300 °C, and crystallites of $SnO_2$ are formed at 450–550 °C.

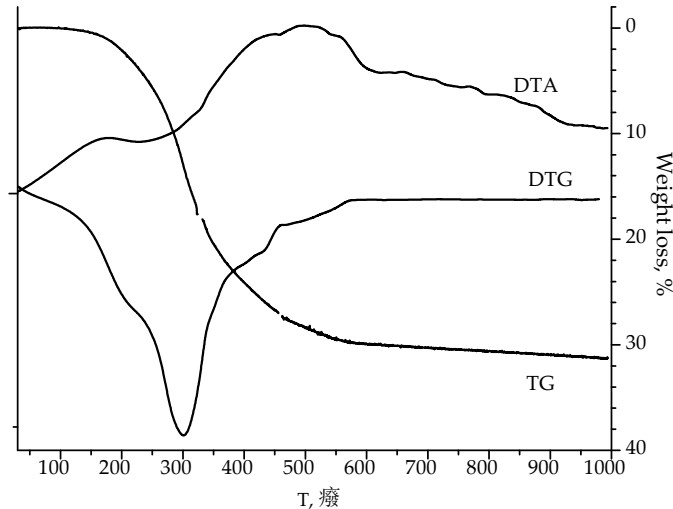

**Figure 1.** TG/DTA curves of as-synthesized $20SnO_2/Al_2O_3$ sample.

The small decrease (10–18%) in the surface area of alumina support occurs upon the deposition of tin oxide what probably explained by blocking of the aluminum pores with the precipitated oxide (Table 1). As seen from the pore size distribution curves (Figure 2), derived from the desorption branches of isotherms using the DFT method, the deposition of $SnO_2$ on alumina surface leads to decrease in pore content with r ≤ 3 nm for $10SnO_2/Al_2O_3$ and $20SnO_2/Al_2O_3$ samples. At the same time, for $25SnO_2/Al_2O_3$ sample, a decrease in pore content with r~4–5 nm is observed. Obviously, the $SnO_2$ crystallites with sizes of more than 3 nm are formed at increase in the content of supported tin dioxide.

The structural analysis by XRD shows that at loadings ≤20 wt.% $SnO_2$ no characteristic peaks assigned to tin oxide were observed (Figure 3). It indicates high dispersion of tin oxide on the alumina surface. Small peaks appeared at $2\theta$ = 26.6°;33.9°;38.0° and 51.8°corresponding to tetragonal $SnO_2$ for $25SnO_2/Al_2O_3$ sample were detected (Figure 3).

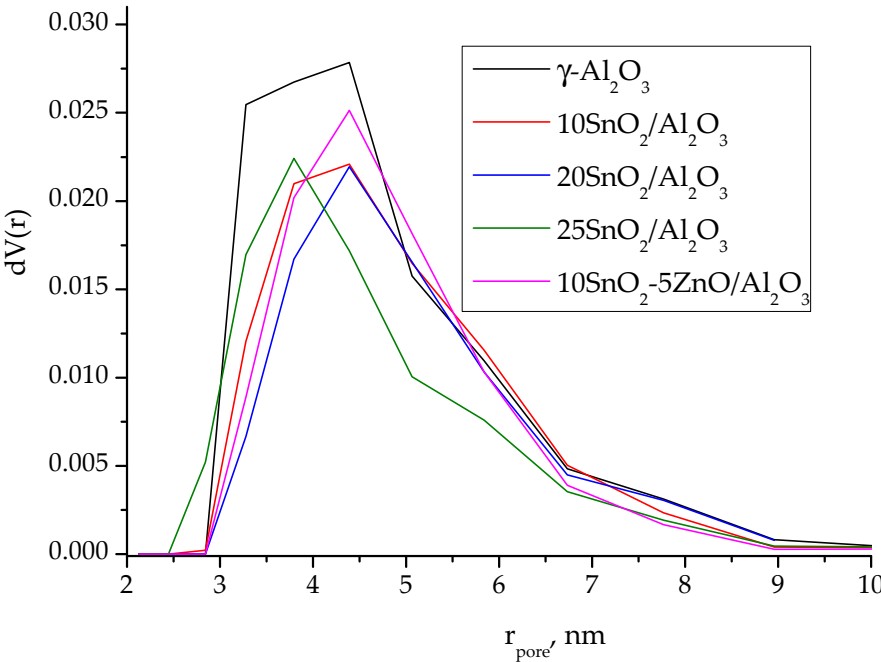

**Figure 2.** DFT pore-size distributions of samples calcined at 550 °C.

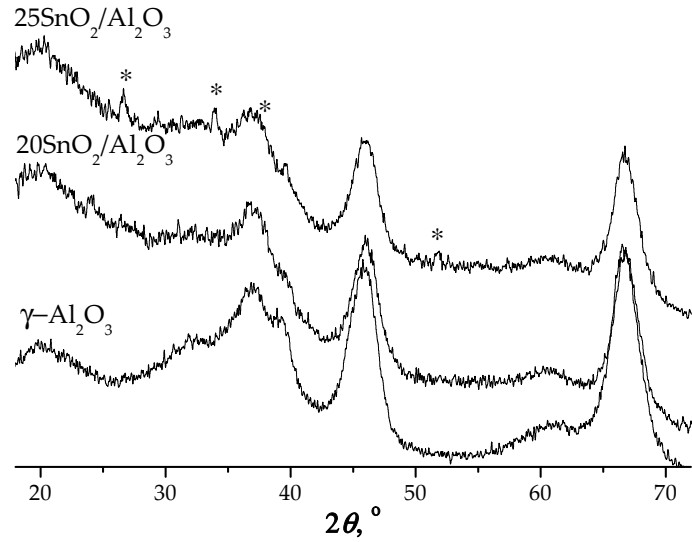

**Figure 3.** XRD patterns of $zSnO_2/Al_2O_3$ samples after calcinations at 550 °C for 2 h.

According to the titration results, $\gamma$-$Al_2O_3$ is weakly acid oxide with $H_0 \leq +3.3$ (Table 1). The addition of $SnO_2$ to alumina surface increases the strength of acid sites of supported $SnO_2/Al_2O_3$ samples to $H_0 \leq +1.5$. Also, at increasing $SnO_2$ content from 10 to 25 wt.%, the concentration of acid sites raises from 1.2 to 1.8 mmol/g (Table 1).

According to the UV-*Vis* data, $^{IV}Sn^{4+}$ or $^{VI}Sn^{4+}$ ions with different coordination are present in studied samples. So, UV−*Vis* spectrum of $25SnO_2/Al_2O_3$ sample shows a broad line around 260 nm, which is attributed to octahedral $^{VI}Sn^{4+}$ ions in $SnO_2$ phase (Figure 4a). However, for $10SnO_2/Al_2O_3$ and $20SnO_2/Al_2O_3$ samples the maximum intensity is observed at 200 nm that attributed to isolated tetrahedral $^{IV}Sn^{4+}$ ions [14] (Figure 4a).

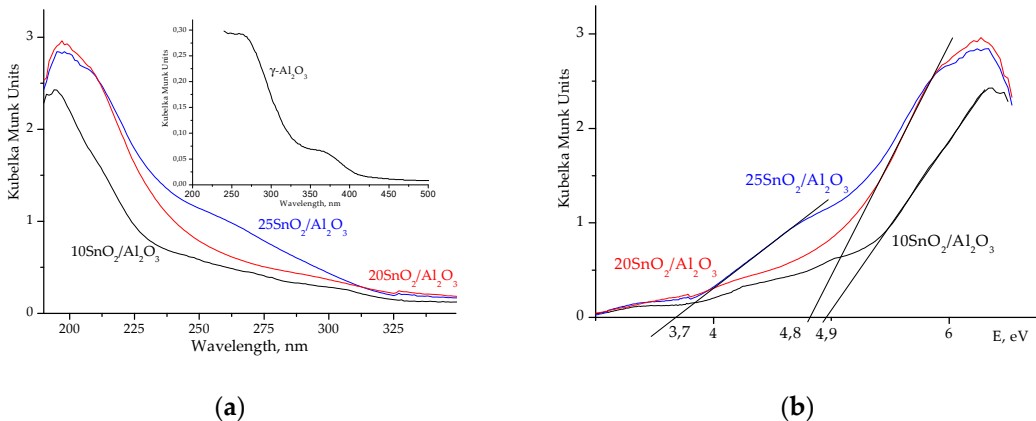

**Figure 4.** UV-Vis diffuse reflectance spectra of $zSnO_2/Al_2O_3$ samples with different Sn loading (**a**); optical energy band gap (**b**).

Figure 4b illustrates the optical band gap ($E_g$) for these samples. It is known [14] that $E_g$ value for massive $SnO_2$ is near 3.55 eV. As stated in [5,16,17], the larger $SnO_2$ particle size, the lower $E_g$ values. The $E_g$ values determined for studied samples are in the interval of 3.7–4.9 eV (Figure 4b) what coincide with similar data in [5,17]. $E_g$ value for γ-$Al_2O_3$ is ~5.25 eV. Thus, for $20SnO_2/Al_2O_3$ sample, the shift of $E_g$ to 4.8 eV indicates the nanosized $SnO_2$ species with $^{IV}Sn^{4+}$ ions that absorb UV light at 200 nm (Figure 4).

Synthesized Sn-containing oxides were tested in the conversion of 13 wt.% fructose-ethanol solution into ethyl lactate. The results are summarized in Table 2. Without a catalyst, the slight dehydration of fructose to 5-hydroxymethylfurfural (5-HMF) is observed at 27% fructose conversion (Table 2).

**Table 2.** Product content of fructose conversion. [1]

| Catalyst | Conversion of Fructose, % | Content of Products, mol% | | | | Yield of EL, % |
|---|---|---|---|---|---|---|
| | | Fructose | EL | 5-HMF | Others [3] | |
| 1    - | 27 | 73 | - | 4 | 23 | - |
| 2    $SnO_2$ | 100 | - | 2 | 85 | 13 | 2 |
| 3    $10SnO_2/Al_2O_3$ | 95 | 5 | 34 | 37 | 24 | 32 |
| 4    $20SnO_2/Al_2O_3$ | 97 | 3 | 51 | 37 | 9 | 49 |
| 5    $25SnO_2/Al_2O_3$ | 95 | 5 | 34 | 58 | 3 | 32 |
| 6    $20SnO_2/Al_2O_3$ [2] | 100 | - | 61 | 34 | 5 | 61 |
| 7    $10SnO_2$-$5ZnO/Al_2O_3$ | 100 | - | 56 | 14 | 30 | 56 |

EL—ethyllactate, 5-HMF—5-hydroxymethylfurfural; [1] Reaction condition: 1,5 g fructose, 0,68 g catalyst, 10 g 98% ethanol, 160 °C, 3 h; [2] 3 mg $K_2CO_3$ was added to 10 g 98% ethanol; [3] Others: β-D-Fructopyranose, benzoic acid, 5-methyl-2-furaldegyde, levulinic and formic acids and its esters.

On pure $SnO_2$ at 160 °C, 100% conversion of fructose occurs with the formation of acid dehydration product—5-hydroxymethylfurfural (Table 2). Levulinic and formic acids and their esters are also formed, as it was in [18].

Target ethyl lactate is formed with 34–53% selectivity at 95–100% fructose conversion on prepared $SnO_2/Al_2O_3$ samples (Table 2). However, the significant amount of 5-hydroxymethylfurfural is also formed (Table 2). For instance, in $^{13}$C NMRspectrum of products obtained on $20SnO_2/Al_2O_3$ at 160 °C, the signals were observed from target ethyl lactate (175.8; 66,9; 61.5; 20.4; 14.2 ppm) and by-products: 5-hydroxymethylfurfural (178.0; 161.4; 152.1; 123.9; 110.1; 57.2 ppm), benzoic acid (172.8; 113.8; 130; 129.4; 128.5 ppm) and 5-methyl-2-furaldegyde (176.8; 159.8; 152; 124; 109.6; 14 ppm) (Figure 5). The intensive ethanol signals at 56; 18.5 ppm and weak lines of unreacted fructose (104; 101.9; 82.8; 81.8; 80.8; 75.7; 75.6; 75.2; 63.6; 62.8; 60.9 ppm) were registered also (Figure 5). The ratio of signal areas at

175.8; 178.0; 172.8; 176.8 ppm allows estimate selectivity of ethyl lactate formation. At that, preliminary the calibration $^{13}$C NMR spectra of ethyl lactate: 5-hydroxymethylfurfural: ethanol mixtures with molar ratios of 0.5:1:10; 1:1:10; 1:2:10 have been recorded.

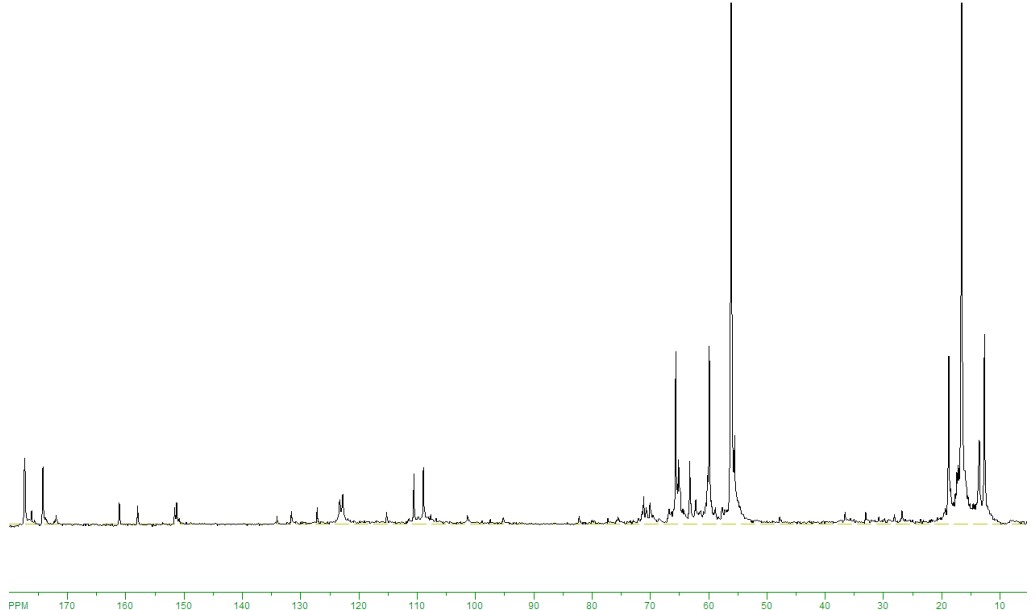

**Figure 5.** $^{13}$C NMR spectrum of fructose conversion products on 20SnO$_2$/Al$_2$O$_3$ (13% fructose solution in ethanol, 160 °C, 3 h; 100 MHz, NS = 256, D0 = 4 s, D1 = 4 μs (30°)).

The addition of a small amount (0.03 wt.%) of potassium carbonate to the fructose–ethanol initial mixture raises the pH value of a reaction solution from 6 to 10. As a result, the yield of ethyl lactate increased due to inhibition of hexose dehydration (Table 2). The same effect was observed in [13]. Ethyl lactate selectivity increased from 53% to 61% at 100% conversion of fructose while simultaneous slight decrease in the amount of dehydration products on 20SnO$_2$/Al$_2$O$_3$ catalyst (Table 2). Further increase in added potassium carbonate up to 0.1 wt.% leads to decrease of ethyl lactate selectivity from 61% to 49%.

We have doped SnO$_2$/Al$_2$O$_3$ with ZnO oxide for decreasing acidity of the catalyst. As a result, ethyl lactate yield increased to 56% with significant decrease of 5-HMF content (Table 2). As seen from Table 1, the highest acid strength (H$_{0max,}$) of this catalyst decreases from +1.5 to +3.3. At that, in comparison with 10SnO$_2$/Al$_2$O$_3$, for 10SnO$_2$–5ZnO/Al$_2$O$_3$ sample the weak base sites with H$_0$ = +7.2 at total content—0.6mmol/g were detected. The decreasing of 5-HMF formation was observed on the Sn-β zeolite doped with Zn$^{2+}$ ions also in [12].

It should be noted that in the experiments we have used a 13% fructose solution in ethanol at mass ratio of fructose/catalyst = 2.4, whereas usually the weaker concentrations of 1–3% with fructose/catalyst = 1.4 were used [8–13].

The principal scheme of fructose–ethanol transformation into ethyl lactate is known [8–10,13,15,16]. The first step of the reaction is the aldol decondensation of fructose to glyceraldehyde and dihydroxyacetone. In acidic medium, the equilibrium shifts to gliceral formation. Further, gliceral dehydration occurs with the formation of pyruval that easily reacts with ethanol to form hemiacetal. Then, there is a subsequent isomerization of hemiacetal into ethyl lactate.

The obtained results show that acid $^{IV}$Sn$^{4+}$ L-sites catalyze aldol decondensation of fructose as the first stage of the ethyl lactate formation (Scheme 1a) as well as initiate the isomerization of hemiacetal into ethyl lactate (Scheme 1b).

**Scheme 1.** Possible schemes of fructose decondensation (**a**) and the rearrangement of hemiacetal into ethyl lactate (**b**) on acid $^{IV}Sn^{4+}$ L-sites.

## 4. Conclusions

Ethyl lactate with 56% yield has been obtained from concentrated 13% fructose solutions in 98% ethanol on supported SnO–ZnO/Al$_2$O$_3$ catalyst at 160 °C. At that, 100% fructose conversion was observed. It is supposed that L-acid $^{IV}Sn^{4+}$ sites provide both C3–C4 aldol decondensation of fructose and subsequent isomerisation of formed hemiacetal of pyruval into ethyl lactate.

**Author Contributions:** Conceptualization, V.V.B.; investigation, S.V.P. and N.L.H.; data curation, S.V.P.; writing—original draft preparation, S.V.P.; writing—review and editing, V.V.B.; project administration, V.V.B.

**Funding:** This research received no external funding

**Conflicts of Interest:** The authors declare no conflicts of interest.

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
