# Peer review of "Conversion of D-Fructose into Ethyl Lactate Over a Supported SnO2–ZnO/Al2O3 Catalyst"

_colloids, doi:10.3390/colloids3010016_

Round 1

Reviewer 1 Report

The paper presents positive results obtained by the authors in the search of simple effective catalysts for D-fructose conversion into ethyl lactate. The samples of g-Al2O3 and the alumina-supported catalysts were adequately studied by different methods. The scheme of transformations taking place during the process seems undoubtable. The results obtained are novel and valuable for the fine organic synthesis.

 Comments:

1. Please, provide the computational error for Ssp determination (2. Materials and Methods, lines 50-61) for the better understanding of “the small decrease in the surface area” (line 87) and “Insignificant decreases of the surface area” (3. Results and Discussion, lines 87 and 101, respectively).

2. Randomly fluctuating data in the case of Ssp changing with the increase in SnO2 content in the catalyst samples must be explained or, at least, noted.

3. The first weakly intense peak assigned to tin oxide detected for 25SnO2/Al2O3 in the Figure 2 is also well-defined for 20SnO2/Al2O3 SnO2. Please, refine your description of the XRD patterns.

4.  In my opinion, the UV-Vis diffuse reflectance spectrum for pure g-Al2O3 must be also shown in Figure 1 for reference.

Author Response

Issue 1

“Please, provide the computational error for Ssp determination (2. Materials and Methods, lines 50-61) for the better understanding of “the small decrease in the surface area” (line 87) and “Insignificant decreases of the surface area” (3. Results and Discussion, lines 87 and 101, respectively).”

Discussion

We agree with this remark.

The corrective line 87 in revised article:

“As seen (Table 1), the small decrease (10–18%) in the surface area of alumina support occurs upon the deposition of tin oxide what probably explained by blocking of the aluminum pores with the precipitated oxide.”

We have deleted line 101. 
Issue 2

“Randomly fluctuating data in the case of Ssp changing with the increase in SnO2 content in the catalyst samples must be explained or, at least, noted”

Discussion

We have extended the explanation in the revised paper. Namely,

“As seen from the pore size distribution curves (Fig. 1), derived from the desorption branches of the isotherms using the DFT method, the deposition of SnO2 on alumina surface leads to  decrease in pore content with r ≤ 3 nm for 10SnO2/Al2O3  and 20SnO2/Al2O3 samples. At the same time, for 25SnO2/Al2O3 sample, decrease in pore content with r ~ 4-7 nm is observed. Obviously, the SnO2 crystallites with sizes of more than 3 nm are formed at increase in the content of supported tin dioxide.”

Issue 3

“The first weakly intense peak assigned to tin oxide detected for 25SnO2/Al2O3 in the Figure 2 is also well-defined for 20SnO2/Al2O3 SnO2. Please, refine your description of the XRD patterns”

Discussion

We do not agree with this remark.

Diffraction patterns were identified by comparing with those from the JCPDS (Joint Committee of Powder Diffraction Standards) data base. Characteristic peaks of SnO2 (Cassiterite) appeared at 2θ = 26.6°, 33.9°, 38.0° and 51.8° only for the 25SnO2/Al2O3 sample. A very small peaks at 2θ = 24.0°, 26.3°, 31° observed for 20SnO2/Al2O3 sample are possibly a recording defect or impurities.

Issue 4

“In my opinion, the UV-Vis diffuse reflectance spectrum for pure γ -Al2O3 must be also shown in Figure 1 for reference”

Discussion

We have recorded the additional UV-Vis spectrum of γ -Al2O3 and have inserted into revised paper. Eg value for γ-Al2O3 is ~5.25 eV.

Thank you,

Dr. Svitlana Prudius

Prof. Volodymyr Brei

Reviewer 2 Report

1)      Regarding detection/quantification of formed products:

-          Concerning 13C NMR characterization, which signals were considered for each product formed? One spectrum with detailed indication must be showed, at least as supplementary material.

-          The parameters of analysis for 13C NMR characterization (number of scans, acquisition time, relaxation delay, etc) must be informed, as well as how the samples are prepared, at least as supplementary material.

-           There is some problems concerning the results presented at Table 2: if the sum of selectivity values is not 100 %, which are the other products formed? As example see entry 1. The authors must clarify this point.

-          Is observed insoluble products formation at these conditions?

2)      In the statement: ……..Further increase in added potassium carbonate up  to 0.1 wt. % leads to decrease of ethyl lactate selectivity from 61 % to 49 %. ……Authors must explain why this result is observed.

3)      In the statement: ……..We have doped SnO2/Al2O3 with ZnO oxide for decreasing acidity of the catalyst……….. What is the effect of the addition of ZnO (as it acts) on the decrease of the acidity of the catalyst? Is acidity actually diminished? this is not clear from the results presented in Table 1. In fact, all the results in Table 1 should be discussed in order to enrich the work.

4)      At Table 1, for 10SnO2–5ZnO/Al2O3, what is the significance of the values between parentheses for Total acidity/basicity and Acid strength?

Sample  

Specific surface area, m2/g

Pore   volume, cm3/g

Average   pore diameter, nm

Total acidity/basicity, mmol/g

Acid   strength H0(H−

10SnO2–5ZnO/Al2O3  

232  

0.65  

10.2  

1.3(0.6)  

≤+3.3(<+7.2)  

Author Response

Thanks

Prof.Brei

Round 2

Reviewer 2 Report

After the suggested corrections, tha article can be acepted.